# Switching and Combining Device-Aided Therapies in Advanced Parkinson’s Disease: A Double Centre Retrospective Study

**DOI:** 10.3390/brainsci12030343

**Published:** 2022-03-02

**Authors:** Dejan Georgiev, Sentilija Delalić, Nina Zupančič Križnar, Achinoam Socher, Tanya Gurevich, Maja Trošt

**Affiliations:** 1Department of Neurology, University Medical Centre Ljubljana, 1000 Ljubljana, Slovenia; senty.d@gmail.com (S.D.); nina.zupancic@kclj.si (N.Z.K.); maja.trost@kclj.si (M.T.); 2Faculty of Computer and Information Sciences, University of Ljubljana, 1000 Ljubljana, Slovenia; 3General Hospital Isola, 6310 Isola, Slovenia; 4Movement Disorders Unit, Tel-Aviv Sourasky Medical Centre, Tel Aviv University, Tel Aviv 69978, Israel; achinoams@tlvmc.gov.il (A.S.); tanyag@tlvmc.gov.il (T.G.); 5Sagol School of Neuroscience, Tel Aviv University, Tel-Aviv 69978, Israel; 6Faculty of Medicine, University of Ljubljana, 1000 Ljubljana, Slovenia

**Keywords:** advanced Parkinson’s disease, deep brain stimulation of the subthalamic nucleus, L-dopa/carbidopa intestinal gel infusion, continuous subcutaneous apomorphine infusion

## Abstract

Background: Device-aided therapies (DAT), such as continuous subcutaneous apomorphine infusion (CSAI), levodopa-carbidopa intestinal gel infusion (LCIG), and deep brain stimulation of the subthalamic nucleus (STN-DBS), have markedly changed the treatment landscape of advanced Parkinson’s disease (aPD). In some patients, it is necessary to switch or combine DATs for various reasons. The aim of this retrospective study was to explore the frequency and reasons for switching between or combining DATs in two movement disorders centres in Slovenia and Israel. Methods: We collected and analysed demographic and clinical data from aPD patients who switched between or combined DATs. Motor and non-motor reasons, adverse events for switching/combining, and their frequency were examined, as was the effect of DAT using the Global Improvement subscale of the Clinical Global Impression Scale, Movement Disorders Society Unified Parkinson’s Disease Rating Scale part III, Mini Mental State Examination, and Parkinson’s Disease Questionnaire 39. Descriptive statistics and non-parametric tests were used to analyse the data. Results: Of 505 aPD patients treated with DATs at both centres between January 2009 and June 2021, we identified in a total of 30 patients (6%) who either switched DAT (*n* = 24: 7 *LCIG-to-STN−DBS*, 1 *LCIG-to-CSAI*, 5 *CSAI-to STN−DBS*, 8 *CSAI-to-LCIG*, 1 *STN−DBS-to-LCIG*, 1 *LCIG-to-CSAI-to-STN−DBS*, and 1 *STN−DBS-to-CSAI-to-LCIG*) or combined DATs (*n* = 6:5 *STN−DBS+LCIG* and 1 *STN−DBS+CSAI-to-STN−DBS+LCIG*). In most of these patients, an inadequate control of motor symptoms was the main reason for switching or combining DATs, but non-motor reasons (related to the disease and/or DAT) were also identified. Conclusions: Switching between and combining DATs is uncommon, but in some patients brings substantial clinical improvement and should be considered in those who have either inadequate symptom control on DAT treatment or have developed DAT-related complications.

## 1. Introduction

Parkinson’s disease (PD) is a chronic, neurodegenerative disease characterized by the loss of dopamine-producing cells in the substantia nigra pars compacta (SNr) [1]. Chronic pulsatile oral treatment with L-dopa and other dopaminergic medication, in addition to neurodegeneration, which plays a major role in the development of late-stage complications in PD, often lead to motor and non-motor complications, such as motor and non-motor fluctuations, and dyskinesias, which in turn affect the health-related quality of life (HRQoL) of patients and their caregivers [1]. Strategies to alleviate these symptoms, such as increasing the frequency and adjusting the L-dopa dosage or switching to, or adding, other dopaminergic medication, may trigger or aggravate dyskinesias on their own merits. Indeed, medication management in patients with advanced PD (aPD) may be challenging and, generally, not very efficient [2].

Device-aided therapies (DAT), namely continuous subcutaneous apomorphine infusion (CSAI), L-dopa-carbidopa intestinal gel infusion (LCIG), and deep brain stimulation of the subthalamic nucleus (STN-DBS), have markedly changed the landscape of aPD in the last decades. Although all DATs improve patients’ motor and non-motor symptoms and HRQoL, they each have certain unique features [3,4]. Several guidelines and algorithms have been published to help neurologists select the most appropriate DAT for each patient [5]. Although the primary goal of all three DATs is to extend ON time without troublesome dyskinesias and improve patients’ HRQoL, each has certain advantages and disadvantages to consider when tailoring the treatment for individuals with aPD. Despite the personalized and multidisciplinary approach that encourages patients and caregivers to actively participate in shared decision making about the selection of DAT, some patients may not experience a satisfactory treatment outcome and may require a switch to another DAT or combining DATs [6]. This can be distressing for patients and caregivers as well as healthcare professionals.

The aim of this study was to investigate the frequency and reasons for switching between or adding DATs in aPD patients in two tertiary movement disorders centres with a long experience with the three DATs: The Centre for Movement Disorders at the Department of Neurology, University Medical Centre Ljubljana, Slovenia, and the Movement Disorders Unit, at the Tel-Aviv Sourasky Medical Centre, Israel.

## 2. Methods

We retrospectively analysed aPD patients who switched between or added additional DAT from January 2009 to June 2021. The total number of aPD patients that started any DAT in both centres was 505: 251 in Ljubljana and 254 in Tel Aviv (Figure 1A). Patients who switched the DAT were categorised into seven subgroups and patients who combined DATs into an additional two: *Switching DATs*: 1. *LCIG-to-STN−DBS*, 2. *LCIG-to-CSAI*, 3. *CSAI-to-STN−DBS*, 4. *CSAI-to-LCIG*, 5. *STN−DBS-to-LCIG*, 6. *LCIG-to-CSAI-to-STN−DBS*, 7. *STN−DBS-to-CSAI-to-LCIG*, and combining *DATs*: 8. *STN−DBS+LCIG* and 9. *STN−DBS+CSAI-to-STN−DBS+LCIG* (Appendix A, Figure 1B). 

In addition to gender, the age at PD onset, the PD duration and age at the introduction of the 1st DAT, the duration of the 1st DAT, the age at the 2nd DAT, the duration of 2nd and 3rd DATs (where applicable), as well as the frequency of the motor and non-motor reasons for switching/combining were recorded and analysed. The Global Improvement (GI) subscale from the Clinical Global Impression scale [7] was used to assess the improvement 6 months after the introduction of DAT. It assesses improvement on a scale from 0 to 7 (0 = Not assessed, 1 = Very much improved, 2 = Much improved, 3 = Minimally improved, 4 = No change, 5 = Minimally worse, 6 = Much worse, and 7 = Very much worse). For the categories *LCIG-to-STN−DBS*, *CSAI-to-STN−DBS*, *CSAI-to-LCIG,* and *STN−DBS+LCIG*, median and range (maximum and minimum) were used as a measure of central tendency. The Mann–Whitney *U* test for two independent samples was used to compare the “switch” to “add-on” groups regarding the above-mentioned variables as well as for the post-hoc analysis for the difference between *LCIG-to-STN−DBS*, *CSAI-to-STN−DBS,* and *CSAI-to-LCIG.* The Kruskal–Wallis *H* test for three or more independent samples was used to compare the switch categories *LCIG-to-STN−DBS*, *CSAI-to-STN−DBS,* and *CSAI-to-LCIG*. For the other categories (*LCIG-to-CSAI*, *STN−DBS-to-LCIG*, *LCIG-to-CSAI-to-STN−DBS*, *STN−DBS-to-CSAI-to-LCIG,* and *STN−DBS+CSAI*-to-*STN−DBS+LCIG*) containing one or two patients each, the data were presented as absolute values only. The Wilcoxon *Z* test was used for the within-group analysis of GI after the first and the second DAT. Data on the motor status as assessed by the Movement Disorders Society Unified Parkinson’s Disease Rating Scale part III (MDS-UPDRS-III), cognitive status as assessed by the Mini Mental State Examination (MMSE), and Parkinson’s Disease Questionnaire 39 (PDQ-39) were available for 5 patients in the *LCIG-to-STN*, 3 patients in the *CSAI-to- STN−DBS*, and 4 patients in the *CSAI-to-LCIG*). These data are presented descriptively in Appendix A. Current state/outcome and reasons for the switch/add-on were also recorded and qualitatively analysed. A *p*-value of less than 0.05 was considered significant. Due to the comparative nature of the study, no correction for multiple comparisons was used. IBM SPSS for MAC v.26.0 was used for analysis. The study was approved by the Medical Ethical Committee of the Republic of Slovenia. 

## 3. Results

The overall number of aPD patients requiring a switch of DAT or additional DAT was 30/505 [(6.0%), 12 females]. Of these, 24/30 aPD patients [(80%), 10 females] switched DAT, and 2 of them switched DAT twice. Additionally, 6/30 [(20%), 2 females] patients needed a combined DAT, and 1 of them switched between two combined DATs (Table 1). There were no significant differences between male and female patients on any of the measured variables (all *p* > 0.201) (Appendix A). 

### 3.1. Reasons for Switching between DATs

*LCIG-to-STN−DBS:* 7/24 (29.1%) patients switched. Unsatisfactory control of motor symptoms was the reason for switching in six of the seven patients: persistent dyskinesias in four, continuous motor fluctuations in three, and freezing of gait along with tremor in one. Additional non-motor reasons and adverse events for switching were found in five of the seven of these patients, and in some there were more than one. LCIG-related polyneuropathy, which was the most common adverse event for the switch to STN-DBS, was observed in three, followed by weight loss in two, punding in one, sleep problems in one, and excessive sweating in one patient.

*LCIG-to-CSAI:* One patient switched due to unsatisfactory control of dyskinesias with LCIG.

*CSAI-to-STN−DBS:* 5/24 (20.8%) switched. Motor reasons were noted in two: the OFF-related dystonia of respiratory muscles and dyskinesias in one patient and the unsatisfactory control of tremors in one. One patient reported bothersome daytime sleepiness and nausea still present one year after CSAI introduction. In one patient, skin nodules at injection sites were the main reason for switching. In two of the five, CSAI was introduced as a “bridging therapy” while waiting for STN-DBS. Although both patients benefited well from CSAI, they preferred to switch to STN-DBS.

*CSAI-to-LCIG:* 8/24 (33.3%) patients switched. Motor reasons (fluctuations in four, dyskinesias in two, freezing of gait in one, and OFF dystonia in one) were noted in seven of the eight. Non-motor reasons were also noted in five of the eight—psychosis in one, visual hallucinations along with sleepiness and fatigue in one, subjective cognitive complaints in one, skin nodules at the injection sites were noted in two patients and skin necrosis in an additional one. In patients in whom motor fluctuations were the reason for change, improvement was observed after switching to LCIG. The age at onset was higher in this group of patients compared to *LCIG-to-STN−DBS* (*p* = 0.002) and *CSAI-to-STN−DBS* (*p* = 0.045). Additionally, the age at the first DAT was higher compared to *LCIG-to-STN−DBS* (*p* = 0.009) and *CSAI-to-STN−DBS (p* = 0.011) as well as the age at the second DAT, *LCIG-to-STN−DBS* (*p* = 0.05) and *CSAI-to-STN−DBS (p* = 0.011) (see Appendix A for group analysis).

*STN−DBS-to-LCIG*: One patient switched because of ongoing severe motor fluctuations and dyskinesias. The DBS was switched off, but not removed. This patient deteriorated relentlessly and deceased 1 year later. 

Two patients switched DATs twice. One patient first switched from *LCIG-to-CSAI* due to severe dyskinesias and autonomic storms during wearing OFF periods. This same patient shortly later switched from *CSAI-to-STN−DBS* because neither dyskinesia nor autonomic storms improved on CSAI treatment. The patient’s condition improved considerably on STN-DBS, and the treatment was efficient for 12 years but then had to be removed due to infection of the DBS system. The other patient switched from *STN−DBS-to-CSAI* because of OFF-related dystonia and cognitive deterioration without any improvement. Two months later the patient was switched to LCIG with partial improvement. Similarly, the DBS was switched off, but not removed. 

There was no difference in any of the groups (*LCIG-to-STN−DBS*, *CSAI-to-STN−DBS* and *CSAI-to-LCIG*) in GI after the first and second DATs (all *p* > 0.316) (Appendix A). Although presented only descriptively, there were no differences in MDS-UPDRS-III, MMSE, and PDQ-39 after the first and second DATs (Appendix A).

### 3.2. Reasons for Combining DATs

Five of six (83.3%) patients had a combination of *STN−DBS* (first) *+ LCIG* (added). In three of these patients, only unilateral STN-DBS was inserted due a predominantly unilateral presentation of the disease in two. In one patient, the unilateral surgery was the first step of a planned bilateral procedure due to the increased risk for cognitive deterioration with mild cognitive impairment and depression, but the second electrode was never inserted. All three patients had continuous motor fluctuations despite unilateral DBS. One of the three mentioned patients with unilateral surgery suffered from a severe form of dopamine dysregulation syndrome (DDS) and a combined LCIG without a satisfactory outcome and was one month later switched back to STN-DBS only. The additional two patients with combined DATs had bilateral STN-DBS. Freezing of gait was the reason for the addition of LCIG in one, and motor fluctuations and dysphagia in the other. 

One patient out of six with combined DAT was first on STN-DBS and had CSAI added because of gait impairment and falls after a period of 10 years of successful STN-DBS treatment. The voltage of stimulation was reduced, and CSAI was introduced with good disease control for another 5 years. After this period, motor fluctuations became difficult to control with this combination, and, instead of CSAI, LCIG was introduced with a favourable control of motor symptoms. Interestingly, urinary retention that appeared when the patient was on *STN−DBS+CSAI* resolved on *STN−DBS+LCIG*. To the best of our knowledge, urinary retention has not been reported as a side effect of CSAI. However, the effect of dopaminergic medication on bladder control is unpredictable, and studies to date have yielded conflicting results [8], with some studies showing that L-dopa and apomorphine improve bladder hyperactivity [9], while other studies show an unpredictable effect [10].

Compared to the group of patients who switched DAT, these patients were older at PD onset (*p* = 0.031), had a shorter PD duration at the first DAT (*p* = 0.025), and had a longer duration of the first DAT before the addition of the second (*p* = 0.020). GI after the first DAT, indicating better satisfaction, was higher in patients who switched DAT (*p* = 0.003). No significant difference was found at GI, either in patients who switched treatment or in patients who combined treatment after the first compared with the second DAT (all *p* > 0.082) (Appendix A).

### 3.3. Motor vs. Non-Motor Reasons for Switching and Combining DATs

Multiple motor and non-motor reasons were identified in most of the patients switching and combining DATs (24/30). Only six patients had a single reason for the change of therapy (Appendix A).

Among motor reasons for switching, persisting motor fluctuations was the most common one for switching (9/24) or combining DATs (5/6) (Figure 2). The second most common cause for switching were dyskinesias (8/24), followed by dystonia (5/24), tremor, and freezing of gait (2/24 each). Less-common causes for double DATs were dyskinesias, dystonia, freezing of gait and gait impairment (1/6 each).

Non-motor causes were much less frequently the reason for switching DATs. Psychiatric and cognitive complains were the reason for switch in four patients, sweating, nausea, and other autonomic nervous system problems in three, sleep problems in three, weight loss in two patients on LCIG, punding in one, and fatigue in another one. Likewise, in patients on combined DAT only two non-motor reasons in two patients, were observed: dysphagia in one and DDS in another. Adverse events, skin nodules in three and skin necrosis in one patient on CSAI and polyneuropathy in three patients on LCIG, were also reasons for switching between therapies. 

## 4. Discussion

Our results show that the overall rate of switching between DATs or combined DATs is rather low at 6.0%, implying that DATs are efficient in the vast majority of aPD patients. Most of the needed switches were from CSAI to either STN-DBS or LCIG, followed by switches from LCIG to either STN-DBS or CSAI. Only two of the patients switched from STN-DBS: one to LCIG and another one to CSAI first and then to LCIG. Six patients were identified with combined therapies, all of whom had STN-DBS as their first DAT. GI 6 months after initiation of the first and second switched or added DAT was similar, arguing that switching or adding DAT is warranted in patients in whom the beneficial effect either wanes or is insufficient. 

### 4.1. Switching between DATs

The age at PD onset and age at 1st and 2nd DAT were highest in patients who switched from CSAI to LCIG and lowest in those who switched from CSAI to STN-DBS. Because CSAI is the least invasive, patients often choose it first. The duration of CSAI treatment is known to be short [11], lasting only a few years, and younger patients intuitively prefer to switch to STN-DBS when needed, while for older patients LCIG is the better option. 

As expected, there was no difference in GI between firstly introduced DATs, confirming a similar effect of all of them. Motor causes (unsatisfactory control of motor fluctuations and dyskinesias) were the most common reason for the switch and similar in all switching combinations. This is not entirely consistent with the results of a recent randomized trial, which showed that LCIG was effective in reducing dyskinesias [12]; however, in our study, the percentage of observed aPD patients who required switching was low. Our results are similar to the findings of a recent study in which 19 patients were sequentially or simultaneously treated with STN-DBS and LCIG [13]. Despite the good initial effect of the first DAT, the recurrence of motor fluctuations and dyskinesias were the most common reason for switching or combining therapies in this Dutch study. Different reasons might have been responsible for the findings in this study, such as younger age of patients at the first DAT, presuming a longer disease duration and the further development of motor complications, higher expectations from the DAT, and unilateral STN-DBS [13]. In our study, most switches were from CSAI to either LCIG or STN-DBS. In a few patients, CSAI was intended as a bridging therapy while patients waited for surgery. These patients had the option to remain on CSAI but preferred to switch. A recent study that directly compared the effects of consecutive treatment with CSAI and STN-DBS [14] showed similar results to ours: STN-DBS had a better overall effect than CSAI, but because CSAI is non-invasive, easy to use, and provides a good benefit for motor symptoms, CSAI may be a convenient treatment option for aPD patients waiting STN-DBS.

OFF-related respiratory dystonia was another observed and already described [15] very disabling motor reason for the switch from CSAI to STN-DBS. As expected, and previously described, skin nodules (and necrosis in one patient) were the main non-motor reason for switching from CSAI to another DAT [11,16,17]. In our study, only one patient switched from CSAI to LCIG due to psychotic symptoms. Interestingly, the age of onset as well as the age at the first and second DATs were the highest in the CSAI-to-LCIG group. Most likely, these patients were not suitable candidates for STN-DBS due to their older age at the time of switching. Consistent with this observation, the CSAI-to STN-DBS group was the youngest with the shortest PD duration before the introduction of DAT.

In our study, most switches from LCIG were to STN-DBS. Again, in most cases, the reasons were the unsatisfactory control of motor fluctuations and dyskinesias. However, polyneuropathy (in addition to the unsatisfactory control of motor symptoms) was the reason for switching from DAT in three patients and weight loss in one patient. Both are known adverse effects of LCIG [18] but may be under-recognised by clinicians in the early years of LCIG treatment. Nowadays, proactive screening for possible polyneuropathy by vitamin B12 blood levels and electromyography is performed in LCIG candidates before and after initiation [19]. 

Interestingly, two patients switched from LCIG to CSAI, both because of uncontrollable motor symptoms. However, one of them switched further from CSAI to STN-DBS in only a few weeks. Inadequate control of motor fluctuations with very severe dyskinesias and autonomic storm that could not be treated with either of the pump-aided treatments improved considerably on STN-DBS in this patient. 

One patient switched from STN-DBS to LCIG because of the recurrence of severe motor fluctuations and dyskinesias after 18 years of successful treatment with STN-DBS. The recurrence of motor fluctuations was probably due to the overall deterioration in this patient that deceased one year after the introduction of LCIG with no improvement of his condition despite the switch. The other patient that switched from STN-DBS first switched to CSAI because of an unsatisfactory control of motor symptoms, in addition to cognitive deterioration. The effect of CSAI was however not sufficient, which lead to the second switch to LCIG with a partial improvement of the symptoms. In both patients, the DBS system was switched off and not removed

### 4.2. Combining DATs

Compared to patients who switched to another therapy, these patients were older at PD onset and had a longer duration of the first DAT before the introduction of the second DAT. Importantly, all the patients who added another DAT were first treated with STN-DBS. Similarly, to the patients who were switching DATs, the unsatisfactory control of motor fluctuations and dyskinesias were the main reasons for the combined DAT in all patients. Importantly, in three of five patients combining STN-DBS with LCIG, unilateral STN-DBS was performed in one because of the increased risk of surgery and in two because of the predominantly unilateral presentation of the disease. This is likely the main reason for the unsatisfactory control of motor symptoms in these patients. However, in one patient, the unsatisfactory control of motor symptoms after years of bilateral STN-DBS stimulation was a reason for adding LCIG. In two of these patients, non-motor symptoms were also identified as an additional cause for double DAT: DDS in one and dysphagia in another. PD duration at first DAT was shorter in patients combining DATs, and the frequency of non-motor reasons for combining was lower than for switching between DATs. In addition, the duration of the first DAT was longer in patients with a combined DAT. This might be explained by the fact that all patients with a combined DAT had STN-DBS as a first DAT, which is known for its long-lasting effect on motor and non-motor symptoms of PD [20]. Interestingly, the improvement after the first DAT in these patients was assessed significantly better (lower GI score) than in patients who were switching between DATs (higher GI score), which might explain the decision to combine, rather than to switch, DATs. 

In only one patient, CSAI was added to STN-DBS. For 10 years, this patient responded very well to STN-DBS, but then developed gait difficulties, falls, and dysarthria. The voltage of the stimulation was reduced, and CSAI was added with a good response for another 5 years. Due to the development of motor fluctuations, CSAI was stopped and LCIG introduced in addition to STN-DBS with a beneficial response.

There are several reports of patients receiving combined *STN−DBS+LCIG* therapy [6,21,22,23], but only one study prospectively investigated the benefit of adding LCIG to STN-DBS in 19 patients [24]. The main reason for the combined DAT therapy in these patients was prolonged OFF time after the introduction of STN-DBS as well as other symptoms refractory to STN-DBS, such as axial symptoms and uncontrollable dyskinesias [6,21,24], due to incorrect patient selection or disease progression requiring high stimulation amplitudes, which can lead to disabling side effects as a result of current spread [6]. In these cases, the addition of LCIG may be regarded as a rescue therapy [23]. In Regidor, Benita, Del Alamo de Pedro, Ley, and Martinez Castrillo [24], out of 19 patients treated by *STN−DBS+LCIG*, 5 patients discontinued combination therapy and returned to DBS monotherapy, 5 discontinued DBS and retained LCIG monotherapy, and 9 patients remained on combination therapy. In five patients who discontinued DBS, there was no clinical deterioration after the implantable pulse generator was exhausted, and it was decided not to replace it. In our patients treated with double DAT, motor fluctuations and disruptive dyskinesias decreased after the introduction of LCIG. In addition, axial symptoms, such as the freezing of gait, improved too. We also observed improvement in some non-motor symptoms such as DDS and dysphagia. 

## 5. Limitations of the Study

This study is a retrospective study, and therefore the results are prone to bias. However, the data collection and its analysis were conducted in two different centres using the same methodology. The assessment of GI was also performed retrospectively. A quantitative motor assessment of patients, as well as PDQ-39 and MMSE, was available only in a limited number of patients and was presented only descriptively. In addition, we did not present data for the patients not switching or combining DATs. However, the main purpose of this study however was to investigate the frequency and causes for switching and combining DATs in PD. The number of patients in the two groups (patients switching DATs and patients with double DATs) was unequal, but the Mann–Whitney *U* test was used to reliably compare these unequal samples [25].

## 6. Conclusions

Switching and combining DATs may be a good option for patients with inadequate symptom control or DAT-related complications. Although the indications/criteria for the introduction of all DATs are similar, this should not prevent the introduction or addition of another DAT. We did not demonstrate a valid difference between switching versus dual DAT choice. Most of our patients switched from CSAI to either LCIG or STN-DBS, in a few cases due to planned bridging therapy while waiting for surgery. Close monitoring of DAT effects and side effects is essential. These should be considered in personalized decision making about the most appropriate type and timing of the first, second, or combined DAT in each individual patient.

## Figures and Tables

**Figure 1 brainsci-12-00343-f001:**
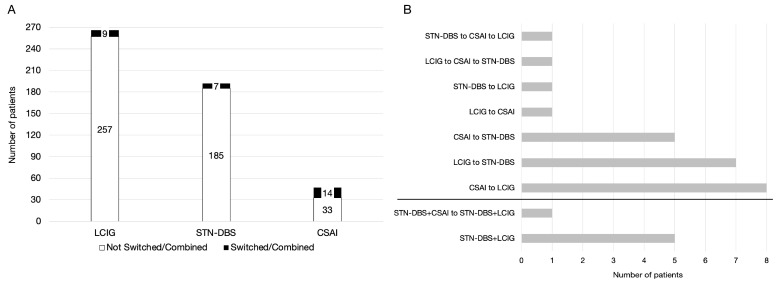
(**A**). Number of patients (*y*-axis) who received continuous subcutaneous apomorphine infusion (CSAI), L-dopa-carbidopa intestinal gel infusion (LCIG), and deep brain stimulation of the subthalamic nucleus (STN-DBS) (*x*-axis) from both centres. (**B**). Number of patients (*x*-axis) on specific switches and combinations of DATs (*y*-axis).

**Figure 2 brainsci-12-00343-f002:**
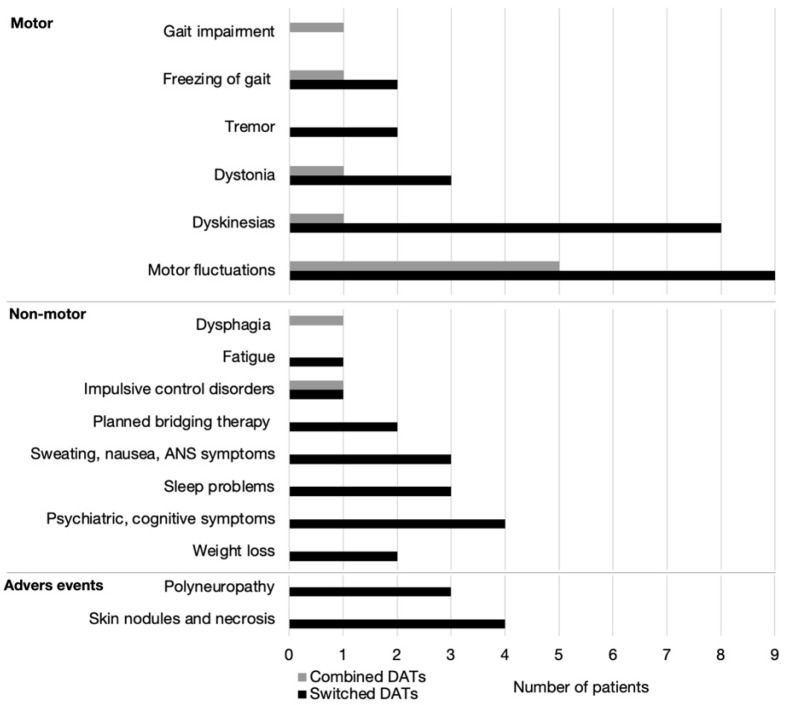
Motor and non-motor reasons and adverse events for switching between (black bars) or combining (gray bars) device-aided therapies (DAT) (*y*-axis). *X*-axis—number of patients.

**Table 1 brainsci-12-00343-t001:** Gender distribution, age at onset of Parkinson’s disease (PD), PD duration at 1st Device-Aided Therapy (DAT), age at 1st DAT, duration of 1st DAT, age at 2nd DAT, and duration of 2nd DAT through June 2021 for patients who switched and patients who combined DAT. Median values and range (in parentheses) are shown. F = females, M = males.

	Switched (*n* = 24)	Combined (*n* = 6)
Gender (F:M)	10:14	2:4
Age at PD onset	49 (29–66)	55 (46–69)
PD duration at 1st DAT	10 (7–28)	7 (3–11)
Age at 1st DAT	60 (38–74)	62 (57–75)
Duration of 1st DAT	1.75 (0.08–18)	5 (3–10)
Age at 2nd DAT	66 (39–77)	68 (64–79)
Duration of 2nd DAT	3 (0.08–9)	3 (0.08–5)

## Data Availability

The data are available in Supplementary Table S1.

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
