# Peer review of "Switching and Combining Device-Aided Therapies in Advanced Parkinson’s Disease: A Double Centre Retrospective Study"

_brainsci, 2022, doi:10.3390/brainsci12030343_

Round 1

Reviewer 1 Report

This is a very comprehensive study that retrospectively analyzes the reasons for switching or combining DAT therapies in two centers. In my opinion, the authors should make some changes in their work:

Abstract: There is no prospective randomized study that compares three DAT, then I suggest to avoid the sentence “ similar outcome of all device-aided therapies”. Retrospective studies showed a very different patient profile for each therapy, specially DBS, this makes not unreliable a direct comparison between them. 

Introduction: not only chronic oral L-dopa leads to motor and non-motor fluctuations

Methods: I found unnecessary an sex-based statistical analysis, since in PD no much differences are related to sex. Maybe this table can be replaced for a “demographical” table, and included in results section.

Results:

  1. Is very hard to read this section, specially sections from 3.1 to 3.3. As the authors comment, the main purpose of this study was to investigate the frequency and causes for switching and combining DAT in PD; then I think that is better provide a descriptive than statistical analysis. As a retrospective study which analyzed high variable reasons to switching a therapy, based on local protocols and individual experience, descriptive analysis is more appropriate. Maybe this material may be move to suppl material. I find more eloquent sections from 3.4 to 3.7.
  2. Related to this sections, I find more interesting figure 2 to understand the reasons to switching, and but not at all what is expressed as F in the tables 1-4. In addition, as the authors comment in limitations paragraph, clinical GI was made in a retrospective manner and motor or QoL scales can not be provided. For all this, I would strongly consider reducing sections from 3.1 to 3.3.
  3. In section 3.4 i suggest not include polineuropathy and skin necrosis as non-motor symptoms, it should be seen as adverse effects. The same for figure 2.
  4. In section 3.5 “combining DAT”, half of patients have a “insufficient DBS” more than a combining DAT. 
  5. Are the authors sure about legend of figure 2 is correct? Black and grey means switched and combined DAT.
  6. I do not know, maybe is printing problem, but size font for tables is not appropriate and line spacing after the tables are not always the same. I suggest, if it is possible, to avoid repeating abbreviations in the legend of each table. 

Discussion and Limitations OK. 

Author Response

This is a very comprehensive study that retrospectively analyzes the reasons for switching or combining DAT therapies in two centers. In my opinion, the authors should make some changes in their work:

RE: We appreciate the reviewer’s comment who finds our study comprehensive.

Abstract: There is no prospective randomized study that compares three DAT, then I suggest to avoid the sentence “ similar outcome of all device-aided therapies”. Retrospective studies showed a very different patient profile for each therapy, specially DBS, this makes not unreliable a direct comparison between them. 

RE: We have changed this sentence. The sentence now reads as follows: “In some patients, it is necessary to switch or combine DATs for various reasons.”

Introduction: not only chronic oral L-dopa leads to motor and non-motor fluctuations

RE: We have changed this sentence. The sentence now reads as follows: “Chronic pulsatile oral treatment with L-dopa and other dopaminergic medication often leads to motor and non-motor complications…”

Methods: I found unnecessary a sex-based statistical analysis, since in PD no much differences are related to sex. Maybe this table can be replaced for a “demographical” table and included in results section.

RE: The section on gender differences is now deleted. Table 1 is now replaced with a table of patients’ demographics. The heading of the table reads as follows: "Table 1. Gender distribution, age at onset of Parkinson's disease (PD), PD duration at 1st Device Aided Therapy (DAT), age at 1st DAT, duration of 1st DAT, age at 2nd DAT, duration of 2nd DAT through June 2021 for patients who switched and patients who combined DAT. Median values and range (in parentheses) are shown."

Results:

1. Is very hard to read this section, specially sections from 3.1 to 3.3. As the authors comment, the main purpose of this study was to investigate the frequency and causes for switching and combining DAT in PD; then I think that is better provide a descriptive than statistical analysis. As a retrospective study which analyzed high variable reasons to switching a therapy, based on local protocols and individual experience, descriptive analysis is more appropriate. Maybe this material may be move to suppl material. I find more eloquent sections from 3.4 to 3.7.

RE: Sections 3.1. to 3.3 are removed from the manuscript. The results from these sections are reduced and briefly mentioned in the new section 3.1. (page 4) “Compared to LCIG-to-STN-DBS and CSAI-to-STN-DBS, age at onset of PD (p=0.011), age at first (p=0.016) and second DAT (p=0.022), and duration of PD (p=0.010) were highest in this patient group (Suppl. Table 3).”;“There was no difference in any of the groups between GI after the first and second DAT (all p>0.316) (Suppl. Table 3).”, and  3.2. (page 5) “Compared to the group of patients who switched DAT, these patients were older at PD onset (p=0.031), had a shorter PD duration at the first DAT (p=0.025), and a longer duration of the first DAT before the addition of the second (p=0.020). GI after the first DAT, indicating better satisfaction, was higher in patients who switched DAT (p=0.003). No significant difference was found at GI, either in patients who switched treatment or in patients who combined treatment after the first comparison with the second DAT (all p> 0.082) (Suppl. Table 4).”

The table on gender differences (old Table 1) is now Supplementary table 2. Old Table 3 is now Supplementary table 3, and Table 2 is now Supplementary table 4. We deleted old table 4 as it does carry important information related to the objective of the study. Consequently, we have also deleted the part referring to it in the Discussion section of the paper: “In addition, there were certain differences between the patients of the two centres, which can be explained by the different structure of switching and combining in the two centres. For example, there were no patients with combined DATs of UMCL and all patients with combined DATs had STN-DBS as their first DAT. Therefore, the differences between the centres are also reflected in the differences between patients switching and combining DATs. The reason for these differences is likely due to local policies regarding the initiation and follow-up of patients with DATs.”

2. Related to this sections, I find more interesting figure 2 to understand the reasons to switching, and but not at all what is expressed as F in the tables 1-4. In addition, as the authors comment in limitations paragraph, clinical GI was made in a retrospective manner and motor or QoL scales can not be provided. For all this, I would strongly consider reducing sections from 3.1 to 3.3.

RE: Please see the reply to the previous comment.

3. In section 3.4 i suggest not include polineuropathy and skin necrosis as non-motor symptoms, it should be seen as adverse effects. The same for figure 2.

RE: Polyneuropathy and skin necrosis are now classified as adverse events in section 3.3. This is also reflected in Figure 2 and throughout the manuscript. For example, at page 6: “Non-motor causes were much less frequently the reason for switching DATs. Psychiatric and cognitive complains were the reason for switch in 4 patients, sweating, nausea, and other autonomic nervous system problems in 3, sleep problems in 3, weight loss in 2 patients on LCIG, punding in 1 and fatigue in another 1. Likewise, in patients on combined DAT only 2 non-motor reasons in 2 patients, were observed: dysphagia in 1 and DDS in another. Adverse events, skin nodules in 3 and skin necrosis in 1 patient on CSAI and  polyneuropathy in 3  patients on LCIG were also reasons for switching be-tween therapies.”

4. In section 3.5 “combining DAT”, half of patients have a “insufficient DBS” more than a combining DAT

RE: This is a good point. This is reflected in the following statement on page 4: “In 3 of these patients, only unilateral STN-DBS was inserted due predominantly unilateral presentation of the disease in 2. In 1, the unilateral surgery was the first step of the planned bilateral procedure due to increased risk for cognitive deterioration in a patient with mild cognitive decline and depression, but the second electrode was never inserted.”, and page 7 “Importantly, in 3/5 patients combining STN-DBS with LCIG, unilateral STN-DBS was performed, in 1 because of increased risk of surgery and in 2 because of predominantly unilateral presentation of the disease. This is likely the main reason for unsatisfactory control of motor symptoms in these patients. However, in 1 patient unsatisfactory control of motor symptoms after years of bilateral STN-DBS stimulation was a reason for adding LCIG.”

5. Are the authors sure about legend of figure 2 is correct? Black and grey means switched and combined DAT.

RE: This is now corrected as follows: Motor, non-motor and adverse events as reasons for switching between (black bars) or combining (gray bars) device added therapies (DAT) (y-axis). X-axis - number of patients.

6. I do not know, maybe is printing problem, but size font for tables is not appropriate and line spacing after the tables are not always the same. I suggest, if it is possible, to avoid repeating abbreviations in the legend of each table. 

RE: This is now corrected in Table 1 as well as in all supplementary tables. The legends of the tables are also rationalised.

Discussion and Limitations OK. 

Reviewer 2 Report

This is the retrospective study exploring the frequency and reasons for switching between or combining device-aided therapy (DAT). The authors concluded that most patients switched from CSAI (continuous subcutaneous apomorphine infusion) to either LCIG (levodopa-carbidopa intestinal gel) or STN-DBS (subthalamic nucleus deep brain stimulation). Switching and combining DATs may be a good option for patients with inadequate symptom control or DAT-related complications. Although the results of this study might be very important for clinical practice, there are major concerns in this manuscript.

The followings are my comment to the author.

  1. As the authors described in the “limitation” section, the major limitations of this study are the lack of results of important clinical scales such as UPDRS, PDQ-39, MMSE/FAB. The severity of motor functions and cognitive impairments is mandatory for determining DAT indications and also for examining switching or combined DAT. Although the authors discussed that the main purpose of this study is to investigate the frequency and causes for switching and combining DAT, motor functions and cognitive functions significantly affect the frequency and cause for switching and combining DAT. Although it might be difficult to obtain the data of UPDRS, MMSE/FAB, and PDQ-39 completely in all patients, however, because this study was conducted in a university hospital, the data of UPDRS, MMSE/FAB, and PDQ-39 might be available in the majority of patients. The readers might be interested in which clinical symptoms are more likely to affect switching or combined DAT in addition to age at PD onset and duration. We recommend the authors present the available data of UPDRS, MMSE/FAB, and PDQ-39 (or PDQ-8).

  1. The readers might be interested in which factors (age, duration, motor, and non-motor symptoms) contribute to switching or combined DAT compared to PD patients receiving single DAT who did not switch or combined.

3 Did the authors perform post-hoc analysis among three subgroups (LCIG to STN-DBS, CSAI to STN-DBS, CSAI to LCIG) of patients switching DAT?

Author Response

This is the retrospective study exploring the frequency and reasons for switching between or combining device-aided therapy (DAT). The authors concluded that most patients switched from CSAI (continuous subcutaneous apomorphine infusion) to either LCIG (levodopa-carbidopa intestinal gel) or STN-DBS (subthalamic nucleus deep brain stimulation). Switching and combining DATs may be a good option for patients with inadequate symptom control or DAT-related complications. Although the results of this study might be very important for clinical practice, there are major concerns in this manuscript.

The followings are my comment to the author.

1. As the authors described in the “limitation” section, the major limitations of this study are the lack of results of important clinical scales such as UPDRS, PDQ-39, MMSE/FAB. The severity of motor functions and cognitive impairments is mandatory for determining DAT indications and also for examining switching or combined DAT. Although the authors discussed that the main purpose of this study is to investigate the frequency and causes for switching and combining DAT, motor functions and cognitive functions significantly affect the frequency and cause for switching and combining DAT. Although it might be difficult to obtain the data of UPDRS, MMSE/FAB, and PDQ-39 completely in all patients, however, because this study was conducted in a university hospital, the data of UPDRS, MMSE/FAB, and PDQ-39 might be available in the majority of patients. The readers might be interested in which clinical symptoms are more likely to affect switching or combined DAT in addition to age at PD onset and duration. We recommend the authors present the available data of UPDRS, MMSE/FAB, and PDQ-39 (or PDQ-8).

RE: We collected some data for the LCIG-to-STN-DBS (5 of 7), CSAI-to-STN-DBS (3 of 5), and CSAI-to-LCIG (4 of 8) groups for UPDRS 3, MMSE, and PDQ39. FAB is not routinely performed in our patients.

There was no difference between groups for UPDRS on the 1st (p=0.519) and 2nd (p=0.723) DAT, MMSE on the 1st (p=0.420) and 2nd (p=0.353) and PDQ -39 on the 1st (p=0.635) and 2nd (p=0.845). At the suggestion of the first reviewer, we decided to delete sections 3.1. through 3.3. and focus on the descriptive statistics, only briefly discussing the quantitative analysis. Because there are no differences for the proposed measures, we would prefer not to include these data, as this would distract the reader from the main goal of the manuscript, which is to retrospectively examine the reasons for switching/combining DAT in a naturalistic setting.

2. The readers might be interested in which factors (age, duration, motor, and non-motor symptoms) contribute to switching or combined DAT compared to PD patients receiving single DAT who did not switch or combined.

RE : This analysis could have yielded very unreliable results because we would have been faced with the problem of comparison between groups with (very) unequal sample sizes. The total number of patients was 505, so we would have compared 30 with 475 patients. Unequal sample sizes lead to unequal variances between samples, which affects the assumption of equal variances in tests such as ANOVA. Both unequal sample sizes and unequal variances dramatically affect statistical power and Type I error rates (Rusticus, S. & Lovato, C., 2014. Impact of Sample Size and variability on the Power and Type I Error Rates of Equivalence Tests: A Simulation Study. Practical Assessment, Research & Evaluation. Vol. 19, No. 11.).

3. Did the authors perform post-hoc analysis among three subgroups (LCIG to STN-DBS, CSAI to STN-DBS, CSAI to LCIG) of patients switching DAT?

RE: Yes, here are the results of the analysis:

Table 1. Post-hoc analysis between LCIG-to-STN-DBS vs. CSAI-to-STN-DBS (for the actual table please see the attachment).

Table 2. Post-hoc analysis between LCIG-to-STN-DBS vs. CSAI-to-LCIG (for the actual table please see the attachment).

Table 3. Post hoc analysis between CSAI-to-STN-DBS vs. CSAI-to-LCIG (for the actual table please see the attachment).

Round 2

Reviewer 1 Report

All my suggestions were properly followed by authors. Only a minor suggestion:

In this sentence "Chronic pulsatile oral treatment with L-dopa and other dopaminergic medication often leads to motor and non-motor complications…" I suggest it would be improve mentioning that neurodegeneration is a key and needed factor to develop PD-complications, not only medication.

Author Response

In this sentence "Chronic pulsatile oral treatment with L-dopa and other dopaminergic medication often leads to motor and non-motor complications…" I suggest it would be improve mentioning that neurodegeneration is a key and needed factor to develop PD-complications, not only medication.

RE: This is now corrected as follows: Page 1. »Chronic pulsatile oral treatment with L-dopa and other dopaminergic medication, in addition to neurodegeneration, which plays a major role in the development of late-stage complications in PD, often leads to motor and non-motor complications....«

Reviewer 2 Report

Thank you very much for revising your manuscript. 1. As the authors described in the “limitation” section, the major limitations of this study are the lack of results of important clinical scales such as UPDRS, PDQ-39, MMSE/FAB. The severity of motor functions and cognitive impairments is mandatory for determining DAT indications and also for examining switching or combined DAT. Although the authors discussed that the main purpose of this study is to investigate the frequency and causes for switching and combining DAT, motor functions and cognitive functions significantly affect the frequency and cause for switching and combining DAT. Although it might be difficult to obtain the data of UPDRS, MMSE/FAB, and PDQ-39 completely in all patients, however, because this study was conducted in a university hospital, the data of UPDRS, MMSE/FAB, and PDQ-39 might be available in the majority of patients. The readers might be interested in which clinical symptoms are more likely to affect switching or combined DAT in addition to age at PD onset and duration. We recommend the authors present the available data of UPDRS, MMSE/FAB, and PDQ-39 (or PDQ-8). RE: We collected some data for the LCIG-to-STN-DBS (5 of 7), CSAI-to-STN-DBS (3 of 5), and CSAI-to-LCIG (4 of 8) groups for UPDRS 3, MMSE, and PDQ39. FAB is not routinely performed in our patients. There was no difference between groups for UPDRS on the 1st (p=0.519) and 2nd (p=0.723) DAT, MMSE on the 1st (p=0.420) and 2nd (p=0.353) and PDQ -39 on the 1st (p=0.635) and 2nd (p=0.845). At the suggestion of the first reviewer, we decided to delete sections 3.1. to 3.3. and focus on the descriptive statistics, only briefly discussing the quantitative analysis. Because there are no differences for the proposed measures, we would prefer not to include these data, as this would distract the reader from the main goal of the manuscript, which is to retrospectively examine the reasons for switching/combining DAT in a naturalistic setting. > We agree that the authors discuss the quantitative analysis only briefly and the purpose of this study is to retrospectively examine the reasons for switching/combining DAT. However, it might be difficult to imagine the clinical characteristics and the severity of PD patients without the data of UPDRS, MMSE, and PDQ-39. Although it might be difficult to perform statistical analysis of UPDSR, MMSE, and PDQ-39 in this study, I recommend the authors to present the available data of UPDRS, MMSE, and PDQ-39. The readers might be able to speculate the relationships between the reasons for switching/combining DAT and PD severity evaluated as the score of UPDRS, MMSE, and PDQ-39. 2. The readers might be interested in which factors (age, duration, motor, and non-motor symptoms) contribute to switching or combined DAT compared to PD patients receiving single DAT who did not switch or combined. RE : This analysis could have yielded very unreliable results because we would have been faced with the problem of comparison between groups with (very) unequal sample sizes. The total number of patients was 505, so we would have compared 30 with 475 patients. Unequal sample sizes lead to unequal variances between samples, which affects the assumption of equal variances in tests such as ANOVA. Both unequal sample sizes and unequal variances dramatically affect statistical power and Type I error rates (Rusticus, S. & Lovato, C., 2014. Impact of Sample Size and variability on the Power and Type I Error Rates of Equivalence Tests: A Simulation Study. Practical Assessment, Research & Evaluation. Vol. 19, No. 11.). > We agree with the authors' comment that statistical analysis comparing the PD patients who received switching/combining DAT and those who did not receive switching/combining DAT is not available due to the unequal sample sizes between two groups. However, the clinical data (age, duration, motor, and non-motor symptoms) of PD patients who did not receive switch/combining DAT might be helpful for readers even though statistical analysis is not available. 3. Did the authors perform post-hoc analysis among three subgroups (LCIG to STN-DBS, CSAI to STN-DBS, CSAI to LCIG) of patients switching DAT? RE: Yes, here are the results of the analysis: > Thank you very much for performing post-hoc analysis among three sub-groups. Do the authors present following table (table 1-3) in this manuscript and discus the results of post-hoc analysis?

Author Response

1. We agree that the authors discuss the quantitative analysis only briefly and the purpose of this study is to retrospectively examine the reasons for switching/combining DAT. However, it might be difficult to imagine the clinical characteristics and the severity of PD patients without the data of UPDRS, MMSE, and PDQ-39. Although it might be difficult to perform statistical analysis of UPDSR, MMSE, and PDQ-39 in this study, I recommend the authors to present the available data of UPDRS, MMSE, and PDQ-39. The readers might be able to speculate the relationships between the reasons for switching/combining DAT and PD severity evaluated as the score of UPDRS, MMSE, and PDQ-39.

RE: The information on MDS-UPDRS part III, MMSE and PDQ-39 is now presented in Supplementary Table 1.

We now indicate this in the Methods section as follows: Page 3 »Data on motor status as assessed by the Movement Disorders Society Unified Parkinson's Disease Rating Scale part III, cognitive status as assessed by the Mini Mental State Ex-amination, and Parkinson’s Disease Questionnaire 39 (PDQ -39) were available for 5 patients in the LCIG-to-STN, 3 patients in the CSAI-to- STN-DBS, and 4 patients in the CSAI-to-LCIG). These data are presented descriptively in Suppl. Table 1.«In the Results section: Page 4 »There was no difference in any of the groups (LCIG-to-STN-DBS, CSAI-to-STN-DBS and CSAI-to-LCIG) in GI after the first and second DAT (all p>0.316) (Suppl. Table 3). Although presented only descriptively, there were no differences in MDS-UPDRS-III, MMSE, and PDQ -39 after the first and second DAT (Suppl. Table 1).«

In the Discussion section: Page 8 »Quantitative motor assessment of patients, as well as PDQ-39 and MMSE was available only in a limited number of patients and was presented only descriptively.«

2. We agree with the authors' comment that statistical analysis comparing the PD patients who received switching/combining DAT and those who did not receive switching/combining DAT is not available due to the unequal sample sizes between two groups. However, the clinical data (age, duration, motor, and non-motor symptoms) of PD patients who did not receive switch/combining DAT might be helpful for readers even though statistical analysis is not available.

 RE: In our opinion the requested quantitive analyses will not add much to the paper, since it is retrospective analyses of the clinical data and not a study planned ahead. On the other hand, there are currently no agreed-upon guidelines/scores for either initiation of device aided therapies (DAT) or switching or combining DATs to "support" our data. Most clinician decisions are based on qualitative assessments and clinicians' personal experiences. We agree that the proposed information may be important to gather more information for the general database on the clinical use of DATs and severity of illness in cases where DAT switch/combination is needed and to objectify these decisions. However, we believe that providing this information is beyond the scope of this paper and would significantly change the concept of the entire paper. We have included this discussion in the limitations section (page 8: »In addition, we did not present data for the patients not switching or combining DATs. However, the main purpose of this study however was to investigate the frequency and causes for switching and combining DATs in PD.«) and request that the manuscript be published in its current form.

3. Thank you very much for performing post-hoc analysis among three sub-groups. Do the authors present following table (table 1-3) in this manuscript and discus the results of post-hoc analysis? 

RE: This is now added in the Methods section: Page 3 »Mann-Whitney U test for two independent samples was used to compare “switch” to “add-on” group regarding the above-mentioned variables as well as for the post-hoc analysis for the difference between LCIG-to-STN-DBS, CSAI-to-STN-DBS and CSAI-to-LCIG«

In the results section, the post-hoc analysis is presented as follows: Page 4 »Age at onset was higher in this group of patients compared to LCIG-to-STN-DBS (p=0.002) and CSAI-to-STN-DBS (p=0.045). Also, age at first DAT was higher compared to LCIG-to-STN-DBS (p=0.009) and CSAI-to-STN-DBS (p=0.011). as well as age at second DAT, LCIG-to-STN-DBS (p=0.05) and CSAI-to-STN-DBS (p=0.011) (see Suppl. Table 3 for group analysis).«